# Lumican/Lumikine Promotes Healing of Corneal Epithelium Debridement by Upregulation of EGFR Ligand Expression via Noncanonical Smad-Independent TGFβ/TBRs Signaling

**DOI:** 10.3390/cells13191599

**Published:** 2024-09-24

**Authors:** Winston W. Y. Kao, Jianhua Zhang, Jhuwala Venkatakrishnan, Shao-Hsuan Chang, Yong Yuan, Osamu Yamanaka, Ying Xia, Tarsis F. Gesteira, Sudhir Verma, Vivien J. Coulson-Thomas, Chia-Yang Liu

**Affiliations:** 1Department of Ophthalmology, University of Cincinnati, Cincinnati, OH 45267, USA; zhanjh@ucmail.uc.edu (J.Z.); venkatja@mail.uc.edu (J.V.); changshao1311@gmail.com (S.-H.C.); yuany@ucmail.uc.edu (Y.Y.); yamanaou@wakayama-med.ac.jp (O.Y.); xiay@ucmail.uc.edu (Y.X.); liucg@ucmail.uc.edu (C.-Y.L.); 2Department of Environmental Health, University of Cincinnati, Cincinnati, OH 45267, USA; 3College of Optometry, University of Houston, Houston, TX 77204, USA; tgferrei@central.uh.edu (T.F.G.); sverma20@central.uh.edu (S.V.); vjcoulso@central.uh.edu (V.J.C.-T.)

**Keywords:** lumican, lumikine, wound healing, cornea, epithelium debridement, TGFβ signaling pathways, EGFR, EGFR ligands

## Abstract

The synthetic peptide of lumican C-terminal 13 amino acids with the cysteine replaced by an alanine, hereafter referred to as lumikine (LumC13_C-A_: YEALRVANEVTLN), binds to TGFβ type I receptor/activin-like kinase5 (TBR1/ALK5) in the activated TGFβ receptor complex to promote corneal epithelial wound healing. The present study aimed to identify the minimum essential amino acid epitope necessary to exert the effects of lumikine via ALK5 and to determine the role of the Y (tyrosine) residue for promoting corneal epithelium wound healing. This study also aimed to determine the signaling pathway(s) triggered by lumican–ALK5 binding. For such, adult *Lum* knockout (*Lum^−/−^*) mice (~8–12 weeks old) were subjected to corneal epithelium debridement using an Agerbrush^®^. The injured eyes were treated with 10 µL eye drops containing 0.3 µM synthetic peptides designed based on the C-terminal region of lumican for 5–6 h. To unveil the downstream signaling pathways involved, inhibitors of the Alk5 and EGFR signaling pathways were co-administered or not. Corneas isolated from the experimental mice were subjected to whole-mount staining and imaged under a ZEISS Observer to determine the distance of epithelium migration. The expression of EGFR ligands was determined following a scratch assay with HTCE (human telomerase-immortalized cornea epithelial cells) in the presence or not of lumikine. Results indicated that shorter LumC-terminal peptides containing EVTLN and substitution of Y with F in lumikine abolishes its capability to promote epithelium migration indicating that Y and EVTLN are essential but insufficient for Lum activity. Lumikine activity is blocked by inhibitors of Alk5, EGFR, and MAPK signaling pathways, while EGF activity is only suppressed by EGFR and MAPK inhibitors. qRT-PCR of scratched HTCE cells cultures treated with lumikine showed upregulated expression of several EGFR ligands including epiregulin (EREG). Treatment with anti-EREG antibodies abolished the effects of lumikine in corneal epithelium debridement healing. The observations suggest that Lum/lumikine binds Alk5 and promotes the noncanonical Smad-independent TGFβ/TBRs signaling pathways during the healing of corneal epithelium debridement.

## 1. Introduction

TGF-β is a major cytokine in tissue morphogenesis, during both embryonic and neonatal development, maintenance of tissue homeostasis, and the repair of diseased tissues in adults. Malfunction of TGF-β signaling in various cell types, e.g., epithelial cells, mesenchymal cells/fibroblasts, and immune cells, has serious consequences, such as epithelium–mesenchyme transition, fibrosis, perturbed immune tolerance, inflammation underlying the pathogenesis of cancer, and scar tissue formation in wound repair [1]. It is of interest to note that there are two major signaling cascades that can be elicited upon the binding of TGFβ to TGFβ receptors: (1) the canonical Smad-dependent pathway and (2) the non-canonical Smad-independent pathway. In the Smad-dependent pathway, the binding of TGFβ to TBR2 (type 2 TGFβ receptor) leads to the phosphorylation of the serine (Ser) and threonine (Thr) residues of TBR2 activating its kinase activity, which subsequently phosphorylates Ser and Thr within the GS domain of ALK5 (TBR1) [2]. The activated TBR1 in the tetrameric TBR2/ALK5 complex then phosphorylates Smad2/Smad3 that binds Smad4 and translocate to the nuclei, where the Smad2/3/4 complex interacts with other transcription factors to activate the expression of target genes. In the noncanonical Smad-independent pathway, the binding of TGFβ to TBR2 leads to the phosphorylation of Tyr residue at the C-terminal domain of TBR2 by Src kinase, which subsequently phosphorylates Y residues in ALK5 and initiates the activation of noncanonical Smad-independent signaling cascades such as ERK, p38MAPK, Rho GTPases and PI3 kinase, etc. [1,3,4]. The signaling cascades elicited by the noncanonical pathway can also be elicited by various other growth factors, cytokines, chemokines, and stress factors. However, the regulatory mechanisms that dictate which TGFβ signaling cascades are elicited upon binding of ligands to TBRs remains largely elusive.

Lumican (Lum) belongs to the small leucine-rich proteoglycans (SLRPs) family. Like most SLRPs, the Lum core protein has a molecular weight of ~40 kDa [5]. The lumican mRNA encodes a pro-lumican molecule that can be divided into four major domains: (1) an N-terminal signal peptide containing 16 hydrophobic-rich amino acid residues; (2) a negatively charged N-terminal domain containing sulfated tyrosine and four conserved cysteine residues forming intra-chain disulfide bond(s); (3) a tandem leucine-rich repeat region (LXXLXLXXNXLSXL)_10_ that binds other extracellular components, e.g., collagen, and cell surface receptors, e.g., integrin; and (4) a C-terminal domain of 53 amino acids consisting two cysteine residues 32 amino acid residues apart that form a conserved intra-chain disulfide bond among vertebrates [5,6,7]. In vertebrates, Lum has a well-conserved core protein that can carry keratan sulfate glycosaminoglycan side chains. Lum core protein is ubiquitously expressed by most cells in connective tissues, if not all. In most connective tissues, it exists as a non-sulfated glycoprotein, except in the corneal stroma where it is one of the three major keratan sulfate proteoglycans (KSPG) besides keratocan and mimican [6,8]. As a constituent of the extracellular matrix (ECM), Lum regulates collagen fibrillogenesis for the formation and maintenance of a transparent corneal stroma, which is evidenced by the cloudy cornea phenotypes seen in *Lum^−/−^* mice [9,10,11]. Lum can also serve as a scaffold and a chemokine (CXCL1) gradient maker for neutrophil extravasation into corneas following injury and inflammation [12,13,14]. Such functions are likely mediated via the interaction of the leucine-rich repeat (LRR) domain with other proteins in the extracellular space and/or cell membrane proteins, e.g., integrin [15]. Interestingly, Lum also attenuates tumor cell migration via binding to integrin receptors [16]. Chakravarti and co-workers also suggested that Lum might interact with Toll-like receptor 4 (TLR4) to mediate its roles in inflammation [17,18,19].

Matrikines are peptides derived from proteolytic degradation of extracellular components, e.g., collagens, fibronectin, laminin, and core proteins of SLRPs, which have cellular functions resembling cytokines and growth factors having pivotal roles in the maintenance of tissue homeostasis and pathogenesis [15,20]. Besides serving as an essential component of the ECM, Lum, as with most SLRP members, possesses multiple matricellular functions during physiological and pathophysiological conditions [21]. For example, both Dcn and Bgn can be internalized via receptor-mediated endocytosis and degraded in lysosomes [22,23,24,25]. This receptor-mediated turnover via endocytosis induced by SLRP members has been suggested as one of the mechanisms by which SLRPs mediate cell signaling regulating inflammation, cell proliferation, and/or apoptosis [21,26], similar to what has been reported for internalization of EGF/EGFR [27], insulin/IGFR, and FGF/FGFR complexes [28]. Recent evidence indicates that the internalization of EGFR following binding to its ligand(s) not only attenuates the EGFR signaling pathway, but also has significant biological consequences in the EGFR signal transduction cascade and modulation of gene expression [29]. Lum also serves multiple matricellular functions such as inhibiting cancer cell growth at early stages of carcinogenesis, e.g., prostate cancer [30], promoting corneal epithelial cell growth and epithelial migration during wound healing and expression of unique corneal proteins, e.g., keratocan, aldehyde dehydrogenase by keratocytes, and altered gene expression patterns by tumor cells [16,31,32,33,34]. As a matrikine, lumican promotes corneal epithelium wound healing and maintains corneal homeostasis by modulating gene expression in normal and diseased tissues [11,13,14,35,36,37]. Maiti, G. et al. reviewed the potential mechanism by which lumican may also participate in modulating inflammation and immune response via internalization of lumican attached to common TLR coreceptors, e.g., CD14 and caveolin 1 [38,39].

Thus, evidence to date suggest there is a cell surface receptor for Lum, which transduces lumican’s matrikine functions [6]. Our previous studies demonstrated that LumC50 peptide_,_ the last C-terminal 50 amino acids domain of Lum, binds to Alk5 of activated tetrameric TGFβR complex to promote the healing of epithelium debridement both in vitro in cultured corneal epithelial cells and in vivo in *Lum*^−/−^ mice [40]. Further, we were able to demonstrate that the last 13 amino acids of lumican, hereafter referred to as lumikine (LumC13_C-A_: YEALRVANEVTLN), was sufficient to promote corneal epithelial wound healing via binding to ALK5 of tetrameric TBR2/ALK5 complex [40,41]. In silico analysis revealed that the LumC-terminal domain interacts with the GS domain of Alk5 within the activated tetrameric Tgfβ receptor complex [41]. Our group was further able to produce highly stable and non-toxic stapled peptides designed based on LumC13 that significantly promote corneal wound healing, providing highly stable and pharmacologically relevant peptides for promoting wound healing [42]. Herein, we identify the minimum/essential LumC13-terminal amino acids for binding to Alk5 to promote corneal epithelial wound healing and determine the role of the Y residue within the Lum C-terminal peptide. It is of interest to note that ablation of Tbr2 in mouse corneal epithelium delays the healing of epithelium debridement, accompanied by a delayed phosphorylation of p38MAPK and ATF2 (activating transcription factor 2) that is essential for epithelium cell migration [43]. This study also elucidates the signaling cascades triggered by LumC13-Alk5 binding that promotes the healing of corneal epithelial debridement.

## 2. Materials and Methods

### 2.1. Animal Maintenance and Corneal Epithelium Debridement Wound In Vivo

Animal care and use conformed to the ARVO Statement for the Use of Animals in Ophthalmic and Vision Research. All animal protocols were approved by the Institutional Animal Care and Use Committee (IACUC) of the University of Cincinnati.

*Lum^−/−^* (*Lum* KO) mice were generated via gene targeting techniques as described previously [9]. Experimental *Lum^−/−^* mice were obtained by cross-breeding male *Lum^+/−^* with female *Lum^−/−^* mice or heterozygous *Lum^+/−^* mice. The mouse colony was outbred with C57BL/6J wild-type mice (Jackson Lab, Bar Harbor, Maine) every 6 months. Both male and female *Lum^−/−^* mice (8~12 weeks old) were anesthetized by intraperitoneal injection of ketamine hydrochloride (5 mg/gm body weight) and xylazine (0.625 mg/gm body weight). The central corneal epithelium was demarcated with a trephine (2 mm in diameter) and subsequently the epithelium was removed using an Agerbrush^®^ (0.5 mm burr, Malvern, PA, USA) under a stereomicroscope, as previously reported [40]. The mice were maintained on a heated pad and eye drops containing LumC peptides (0.3 µM) and/or inhibitors in PBS were applied as eye drops every 10 min for 4–6 h. To identify the minimum and essential lumican C-terminal peptides that promote the healing of corneal epithelium debridement, a series of C-terminal peptides, i.e., LumC4, LumC5, LumC7, LumC10, LumC13, LumC13_C-A_ (lumikine), Lumikine_Y-F_ (LumC13_C-A/Y-F_), LumC-hybrid1/3, LumC-hybrid2/3, LumC-hybrid2/3_Y-F_, LumC33_∆C20_, and LumC18_∆C5_, were synthesized (New England Peptide, Gardner, MA, USA). Appendix A shows the amino acids sequences of LumC peptides. In order to elucidate the downstream signaling cascades, individual peptides were co-administered or not with EGF (10 ng/mL) in PBS plus cytokine and MAPK inhibitors, i.e., ALK5 inhibitor SB431542 (10 µM); EGFR inhibitor AG1478 (10 nM); pERK1/2 inhibitor (PD98059, 5 µM); PI3K inhibitor (Wortmannin, 1 µM); and Src inhibitor (SrcI-1, 2 µM), according to the manufacturers’ instructions (All inhibitors were purchased from R & D, Minneapolis, MN, USA). Mice were euthanized by ketamine/xylazine injection followed by cervical dislocation at either 4, 5, or 6 h and eyeballs were collected and fixed in 4% paraformaldehyde (PFA) in PBS at room temperature for 30 min. Corneas were isolated, quenched with 0.1% NaBH4 and processed for wholemount staining with phalloidin and DAPI at 4 °C overnight. Images were taken with a Zeiss Apo Tome microscope (Observer Z1, Zeiss Oberkochen, Baden-Württemberg, Germany). The removal of epithelium leads to a slight elevation of stroma beginning at the wound edge due to hydration caused by the loss of epithelium. The front moving distance is measured between the migrating tip and origin of wound edge as shown in Figure 1A–D.

### 2.2. Quantitative Real Time PCR (qRT-PCR) of EGFR Ligands

Sub-confluent (80–90%) human telomerase-immortalized corneal epithelial (HTCE) cells were grown on 8 × P100 plates in complete media (Dermalife, Lifeline Cell Technology, Frederick, MD, USA). At confluence, complete media were changed to basal media (without growth factors) to starve the cells overnight. The next morning, 5 linear parallel equidistant scratches were made using P200 microtip and plates washed 3X with PBS to remove any loose cells/debris. Fresh basal media containing or not 0.3 µM lumikine (LumC13_C-A_) were added and incubated at 37 °C in 5% CO_2_. A vehicle control (basal media) was run in parallel. The media were collected and stored at −80 °C after 30 min (treatment 1) and 2 h of incubation (treatment 2), and Trizol (Invitrogen, Carlsbad, CA, USA) was added to the cells. Total RNA was extracted to synthesize cDNA and quantitative real time PCR was carried out using specific primers as listed in Appendix A. Briefly, total RNA was isolated from the cells using Trizol Reagent (Invitrogen, Carlsbad, CA, USA) according to the manufacturer’s instructions. RNA concentration and purity were determined using a nanodrop spectrophotometer (NanoDrop 2000, ThermoFisher Scientific, Waltham, MA, USA) at 260 and 280 nm; 2 μg of total RNA was used for synthesizing first-strand cDNA with the high-capacity cDNA reverse transcription kit (ThermoFisher, Waltham, MA, USA), according to the manufacturer’s instructions. The expression levels of genes of interest were analyzed by quantitative real-time PCR amplification (qPCR) using a Powerup SYBR Green Master Mix kit (ThermoFisher, Waltham, MA, USA) using a BIORAD CFX Connect Real-time System. Amplification consisted of an activation cycle of 95 °C for 10 min followed by 40 cycles of 95 °C for 15 s and 60 °C for 1 min. Relative quantification of the expression levels was carried out against two housekeeping genes, Gapdh and β-actin, using both the 2^−ΔCt^ and 2^−ΔΔCt^ method.

### 2.3. Immunofluorescence Staining for EGFR-1068 and -1069-Phosphorylated Tyrosine

Experimental *Lum^−/−^* mice were injured and treated with eyedrops containing lumikine (0.3 µM in PBS) or PBS only for 6 h and euthanized, as described above. The enucleated eyes were fixed in 4% paraformaldehyde in PBS at 4 °C overnight and were then subjected to cryosections and immunofluorescence staining with rabbit antibodies against EGFR (MilliporeSigma, Burlinton, MA, USA, Cat# 06-847), anti-EGFR-pho-1068Y (MilliporeSigma, Burlinton, MA, USA, Cat# 04-339), and anti-EGFR-Pho-1069 Y (MilliporeSigma, Burlinton, MA, USA, Cat#09-310), as described previously [44].

### 2.4. Inhibition of Lumikine Effects on Epithelium Wound Healing by Anti-TGFβ and Anti-EREG Antibodies

Mouse monoclonal-neutralizing anti-human TGFβ 1,2,3 antibody (CAT# MAB 1835) cross-reacts with humans, mice, chickens, etc., and goat anti-mouse EREG (CAT#AF1068) was purchased from R & D (Minneapolis, MN, USA). The antibodies were included in the eyedrops containing lumikine (0.3 µM), mouse monoclonal anti-human TGFβ 1,2,3 (10 µg/mL), and goat anti-mouse EREG (20 µg/mL). The eye drops were applied to the injured corneas as described above.

### 2.5. Statistical Analysis

Data are presented as mean ± standard deviation (SD). One-way ANOVA followed by Sidak’s multiple comparison test was performed to determine the statistical significance. To assess the expression of EGFR ligands by HTCE cells treated with lumikine, we conducted a two-way ANOVA followed by Sidak’s multiple comparison test. Statistical analysis was performed using GraphPad Prism version 10.1 (GraphPad Software, Inc., San Diego, CA, USA). *p* value is indicated as follows: * *p* < 0.05, ** *p* < 0.01, *** *p* < 0.001, **** *p* < 0.0001.

## 3. Results

### 3.1. Effects of Antibody against TGFβ on the Activity of Lumikine (LumC13_C-A_) in the Healing of Corneal Epithelium Debridement (CED)

Our previous studies found that in vitro GST (glutathione S-transferase)-LumC50 fusion protein pulldown assay by immune precipitation and Western blot analysis showed that LumC-terminal domain binds to Alk5 [40]. In silico analysis revealed that the LumC-terminal domain (LumC50) interacts with the GS domain of Alk5 within the activated tetrameric Tgfβ receptors complex [41]. Further, we were able to demonstrate that the last 13 amino acids of lumican LumC13 (YECLRVANEVTLN) was able to promote corneal epithelium wound healing in the presence of a reducing agent such as 2-mercaptoethnol. The observation suggests that it was the dimerization of LumC13 via the disulfide bond between two LumC13 molecules that blocked the LumC13 effect on epithelium wound healing. Therefore, a peptide named lumikine (LumC13_C-A_: YEALRVANEVTLN) was synthesized in which C is substituted with A (as shown in Appendix A) and used to verify that lumikine can indeed account for the Lum matrikine functions in promoting corneal epithelial wound healing.

To verify whether TGFβ signaling is involved in the enhanced healing of corneal epithelium debridement (CED) by lumikine and to establish a protocol for quantifying the healing rate of CED, anti-TGFβ antibodies were added or not to eye drops containing lumikine (0.3 µM) and EGF (10 ng/mL). Epithelial migration rate was measured with a ZEISS inverted apotome microscope as described in Materials and Methods. Epithelial migration is measured from wound edge to the tip of migrating epithelium over the elevated stroma that is caused by stroma hydration following the loss of epithelium. Figure 1A demonstrates that in the absence of antibodies, lumikine promotes the epithelium migration (~123 µm). In contrast, the addition of anti-TGFβ antibody abolishes the effect of lumikine on epithelial migration of CED (~24 µm, Figure 1B). In contrast, anti-TGFβ antibody does not abolish the enhanced epithelial migration by EGF, comparing Figure 1C (~115 µm) to Figure 1D (~102 µm). There is a minimal epithelial migration of ~25 µm as shown in Figure 2 in corneas treated with PBS alone. Figure 2 shows the cumulative data indicating that anti-TGFβ antibodies inhibit the effects of lumikine in the healing of CED but has no effects on EGF promoted healing of CED. The anti-TGFβ antibody has no statistically significant effect on the healing of CED treated with eye drops of PBS alone. This observation indicates that either EGF signaling is downstream of Lum/Tgfβ/Tbrs signaling, or that EGF/EGFR signaling is independent of Lum/Tgfβ/Tbrs signaling. Further experiments using inhibitors of Alk5, EGFR, and MAPK were performed to validate these hypotheses, as described below.

### 3.2. Identification of the Minimum/Essential LumC-Terminal Domain for Promoting Corneal Epithelium Wound Healing

A series of Lum C-terminal peptides were synthesized (as shown in Appendix A) and used to identify the minimum essential and sufficient LumC peptides that can promote the healing of CED. The peptides were tested in vivo using epithelium debridement with *Lum^−/−^* mice as an experimental model. As shown in Figure 3, the peptide LumC18_∆C5_ (LPPDMYECLRVAN) missing the last 5 C-terminal amino acids “EVTLN” failed to promote corneal epithelium wound healing. Other shorter C-terminal peptides containing EVTLN, e.g., LumC10, LumC7, LumC5, and LumC4 presented a much lower capability to promote epithelial wound healing when compared to that of lumikine. Interestingly, LumC5 presented a slightly higher capacity to promote corneal epithelial migration than those of LumC10 and LumC4, not achieving significance for LumC7. These observations are consistent with our previous findings that lumikine derived from LumC13 is the minimal essential and sufficient LumC peptide for promoting corneal epithelial wound healing [40,41], with the last 5 C-terminal amino acids (EVTLN) being essential for lumikine in promoting the healing of debrided corneal epithelium.

### 3.3. Identification of the Y Residue as Integral for Lumikine in the Healing of CED

Our previous in silico analysis and in vivo wound healing studies suggested that the Y residue at the N-terminal end of lumikine participates in the formation of a hydrogen bond with the GS domain of Alk5 [41]. To further validate the observation that the Y residue of lumikine is indeed pivotal for promoting corneal epithelial wound healing, a series of peptides were designed and synthesized. These included a peptide where Y of LumC13_C-A_ was substituted with F (Lumikine_Y-F,_ LumC13_C-A/Y-F_), hybrid peptides containing C-terminal EVTLN and N-terminal sequences within LumC50 either containing Y Hybrid2/3, an F substitution of Hybrid2/3_Y-F_ and Hybrid1/3 (lacking Y), as shown in Figure 4. The substitution of Y with F in Lumikine_Y-F_ and Hybrid2/3_Y-F_ lack the capability to promote epithelial wound healing, confirming the in silico data that the formation of hydrogen bond between Y of LumC13 and Alk5 GS domain play a pivotal role in corneal epithelium wound healing [41].

### 3.4. Identification of Downstream Signaling Pathway(s) Triggered by Lum/Lumikine

As mentioned above, the co-administration of anti-TGFβ neutralizing antibody to the ocular surface following corneal epithelium debridement abolished the effects of lumikine in promoting corneal epithelial wound healing; however, it did not affect the ability of EGF to promote wound healing (Figure 1 and Figure 2). This suggests that either the EGF/EGFR signaling is downstream of lumikine/TGFβ/TBRs signaling, or they are two independent parallel pathways. To determine which of these is true, a series of experiments were carried out using several inhibitors of the TGFβ and EGF signaling pathways. For such, a series of debridement wounds were made on *Lum^−/−^* mice, lumikine (0.3 µM) or EGF (10 ng/mL) was administered to the ocular surface via eye drops together with either ALK5 inhibitor SB431542 (10 µM), EGFR inhibitor AG1478 (10 nM), pERK1/2 inhibitor (PD98059, 5 µM), PI3K inhibitor (Wortmannin, 1 µM), and Src inhibitor (SrcI-1, 2 µM). PBS only was used as the control. Figure 5A shows that in the absence of inhibitors both lumikine and EGF promote healing of epithelium debridement. All inhibitors tested significantly inhibited corneal epithelial wound healing when compared to PBS alone, with PI3K having the most pronounced effect (Figure 5B). When the inhibitors were co-administered with lumikine, they all significantly inhibited wound healing when compared to lumikine alone, thereby preventing lumikine from promoting wound healing (Figure 5C). In contrast, when inhibitors were co-administered with EGF, the ALK5 inhibitor SB431542 and Src inhibitor (SrcI-1) did not affect the ability of EGF to promote wound healing, while EGFR inhibitor (AG1478), pERK1/2 inhibitor (PD98059), and PI3K inhibitor (wortmannin) did inhibit the ability of EGF to promote wound healing (Figure 5D). This observation is consistent with notion that the interaction of lumikine and Alk5 may have a role in promoting the noncanonical Smad-independent TGFβ signaling pathway that involves MAPK signaling pathways. To validate this possibility, immunofluorescence staining with antibodies against EGFR phospho-1068Y and -1069Y was performed, demonstrating that lumikine treatment results in the phosphorylation of 1068Y and 1069Y in activated EGFR, suggesting that binding of lumikine to Alk5 triggers TGFβ/TBRs signaling that may facilitate the noncanonical Smad-independent TGFβ signaling pathway (Figure 6). The observation is consistent with the notion that the effects of lumikine on the healing of CED may be mediated via activation of EGFR signaling. For example, it is highly likely that administration of lumikine may enhance the expression of EGFR ligands by epithelial cells, which may then feed forward to activate the TGFβ-noncanonical Smad-independent signaling pathways for the healing of CED.

### 3.5. Effects of Lumikine on the Expression of EGFR Ligands

To assess the effects of lumikine on EGF signaling, cultured confluent HTCE cells were subjected to a scratch wound and immediately treated with media containing lumikine (0.3 µM) for either 30 min or 2 h, and cells without lumikine treatment were used as control. Total RNAs were isolated and analyzed by quantitative RT-PCR with primers for various EGFR ligands as shown in Appendix A. Within the first 30 min of treatment, no significant changes in the expression of EGFR ligands were observed, i.e., HBEGF (heparin-binding EGF), TGFα, EREG (epiregulin), BTC (betacellulin), EGF, and AREG (amphiregulin) (Figure 7 and Appendix A). However, after 2 h there was a significant increase in the expression levels of several EGFR ligands, namely, HBEGF, TGFα, EREG, BTC, EGF, and AREG, with TGFα, EREG, and AREG having the most pronounced increase in expression. The expression levels of BTC and EGF are much lower than other ligands examined; therefore, they may not have a significant role in mediating the effects of lumikine on promoting corneal epithelium wound healing. In further experiments, the role of EREG on epithelium wound healing was studied in vivo using anti-ERGE antibodies. The administration of anti-EREG neutralizing antibodies (20 µg/mL) abolished lumikine effects on corneal epithelium wound healing in vivo (Figure 8). These observations are consistent with the hypothesis that EGFR signaling is downstream of Lum/TGFβ/TBRs signaling cascades in the healing of corneal epithelium debridement. The binding of lumican/lumikine to TBRs enhances the upregulated expression of EGFR ligands that enhance the TGFβ noncanonical Smad-independent pathway during the healing of CED.

## 4. Discussion

Our previous studies demonstrated that in the absence of lumican, there is a delay of corneal epithelium wound healing in *Lum^−/−^* mice [9]. This defect in epithelium debridement healing could be improved by the administration of GST-LumC50 fusion protein and lumikine [40]. In the present study, we confirmed the minimal essential and sufficient lumikine (LumC13_C-A_) derived from the last C-terminal 13 amino acids of lumican core protein for promoting corneal epithelium wound healing. The last five C-terminal amino acids EVTLN of lumican is essential but not sufficient for optimal lumikine (YEALRVANEVTLN) functions (as shown in Figure 3 and Figure 4). We were also able to identify that the Y residue of LumC13 is essential for promoting corneal epithelium wound healing. An observation that is consistent with our previous in silico analysis indicating that the Y residue is necessary for the binding of lumican to Alk5 via establishing a hydrogen bond with the GS domain of Alk5 [41,42].

It is well known that TGFβ signaling has a pivotal role in the transition of fibroblasts to myofibroblasts leading to fibrosis in diseased organs and scar tissues formation of injured tissues [4,45]. TGFβ also has pivotal roles in corneal wound healing [2]. In the present study, we demonstrated that the administration of anti-TGFβ antibody abolished the effects of lumikine, but not that of EGF, on the healing of corneal epithelium debridement in *Lum^−/−^* mice (Figure 1 and Figure 2). This observation suggests that either EGF signaling is downstream of TGFβ signaling cascades, or that they are two independent parallel signaling pathways. Our previous studies demonstrated that TGFβ signaling promotes the healing of corneal epithelium debridement via the activation of phospho-p38MAPK instead of the canonical Smad-dependent TGFβ signaling cascades [46]. Interestingly, Hutcheon et al. demonstrated that corneal superficial keratectomy involves epithelium debridement; plus, the removal of basement membrane underlying epithelium underwent characteristic nuclear translocation of Smad2/3/4 complex, suggesting that it involved the canonical Smad-dependent TGFβ/TBRs signaling cascades [47]. Further, the ablation of TGFβ type 2 receptor in corneal epithelium hampers the healing of corneal epithelium debridement accompanied by a delay of phospho-p38 MAPK activation [43].

In general, the binding of TGFβ to TBR2 initiates the autophosphorylation of TBR2 and activates its kinase activity that subsequently phosphorylates Ser and The residues of ALK5 (TBR1) in the tetrameric TBRs complex. The active tetrameric TBRs complex phosphorylates Smad 2 and 3 that then binds Smad4 to form a Smad 2/3/4 complex that then translocate into nuclei, where it binds other transcription factors and drives the expression of TGFβ target genes, including components of extracellular components. In noncanonical Smad-independent pathways, it has been shown that SRC binds free TBR2; upon TGFβ binding to TBR2, SRC phosphorylates 284Y of TBR2 and forms tetrameric TBRs that phosphorylate Alk5 Y residues and transduce the MAPK signaling cascades [48,49]. However, the molecular mechanisms involved remain elusive, for example, the molecular mechanism(s) that make the cells trigger Smad-dependent or Smad-independent signaling cascades during the epithelium wound healing in wild type mice. The data in this present study demonstrate that the effects of lumikine on corneal epithelial wound healing is abolished by inhibitors of Alk1, Src-1, EGFR, Erk, and PI3K. In contrast, the effect of EGF on corneal epithelium wound healing was not affected by Alk5 and Src-1 inhibitors, but by EGFR, Erk, and PI3K inhibitors, as shown in Figure 5. Interestingly, the addition of lumikine enhanced the phosphorylation of EGFR, as shown in Figure 6. These observations suggest that EGFR signaling is downstream of TGFβ/TBRs signaling via the activation of noncanonical Smad-independent TGFβ signaling cascade. The suggestion is supported by the upregulated expression of EGFR ligands, e.g., HBEGF, TGFα, EREG, AREG, BTC, and EGF. In an in vitro wound healing model of scratched HTCE cell cultures, EREG is upregulated in the presence of lumikine, as shown in Figure 7. This finding is consistent with previous studies by Draper et al. in that EGFR ligands promote the healing of a murine excision wound model [50]. Further, the ablation of EREG in KO (*EREG^−/−^*) mice leads to enhanced corneal inflammation following epithelium debridement [51]. In further experiments, we found that the administration of anti-EREG antibodies abolished the effects of lumikine on corneal epithelial wound healing, as seen in *Lum^−/−^* mice (Figure 8). Taken together, the data suggest that the upregulated expression of EGFR ligands exerts a feedforward mechanism to amplify the EGFR signaling of p38 MAPK and JNK signaling cascades, thereby improving corneal epithelium wound healing. Thus, the results of our current study suggest that lumikine plays a pivotal role in triggering the noncanonical Smad-independent TGFβ/TBR signaling cascade observed in the healing of corneal epithelium debridement. Figure 9 summarizes our hypothesis on how lumican and lumikine dictate the noncanonical Smad-independent TGFβ signaling cascades in the healing of corneal epithelium debridement. Namely, upon the binding of TGFβs to TBR2/SRC complex, the SRC kinase phosphorylates 284Y of TBR2; then, the binding of lumikine to Alk5 leads to the Y phosphorylation in Alk5 and, consequently, upregulates the expression of EGFR ligands that subsequently activate p38 MAPK/JNK signaling for cell proliferation and cell migration during corneal wound healing. Thus, the presence of lumikine may serve as a switch to turn on the noncanonical Smad-independent TGFβ/TBRs signaling cascades, e.g., p38 MAPK signaling pathways during the healing of corneal epithelium debridement. It is worthy to note that upon wounding, the availability of free lumican molecules may be a result from the expression of lumican by corneal epithelial cells and keratocytes of the injured cornea, or from the release of lumican via proteolysis of the stroma extracellular matrix following corneal injury [13,43,46].

It is of interest to note that the healing of corneal superficial keratectomy that involves epithelium debridement plus the removal of basement membrane underlying epithelium is characterized by the nuclear translocation of Smad2/3/4 complex, suggesting that it involves the canonical Smad-dependent TGFβ/TBRs signaling cascades [47]. In contrast, epithelium debridement undergoes the TGFβ noncanonical Smad-independent signaling cascade as shown in Figure 9. The exact molecular mechanisms that regulate the type of TGFβ signaling cascades in cornea epithelium debridement (CED) and cornea superficial keratectomy (CSK) are not known. However, it may be related to the exposure of corneal epithelial cells to different extracellular matrixes, such as the basement membrane in CED and stromal collagenous matrices in CSK. Alternatively, it is likely that the strength of TGFβ signaling corresponds to the availability of active TGFβ released from latent TGFβ in ECM in either type of injured corneas, such as there being more active TGFβ in CSK than that of CED. Further studies are needed to determine the molecular and cellular mechanisms that may govern the TGFβ signaling cascades in the healing of corneal superficial keratectomy that often results in corneal scar tissue formation. In addition, further studies are also needed to determine whether the administration of lumican/lumikine would ameliorate the pathology in tissue fibrosis, e.g., liver and lung, and prevent scar tissue formation in injured tissues such as corneal alkaline burn, laceration, keratectomy, etc.

## Figures and Tables

**Figure 1 cells-13-01599-f001:**
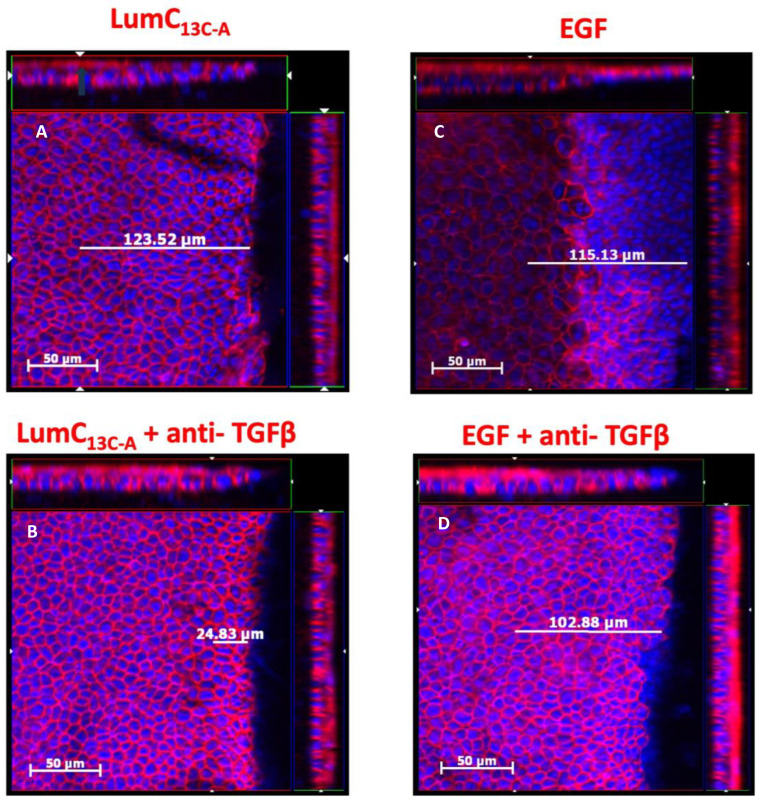
Effects of anti-TGFβ antibodies on lumikine and EGF promoted the healing of cornea epithelium debridement. Adult 8–12-week-old *Lum^−/−^* KO mice were subjected to epithelium debridement with Algerbrush^®^. The injured corneas were treated with 10 µL PBS eyedrops containing 0.3 µM lumikine (**A**), lumikine + Anti-TGFβ antibodies (10 µg/mL) (**B**), EGF (10 ng/mL) (**C**), and EGF + anti-TGFβ (10 µg/mL) (**D**) every 10 min for 3 h. Experimental mice were sacrificed and excised eyes were fixed in 4% PFA/PBS at room temperature for 1 h and then quenched with 0.1% NaBH_4_. Dissected corneas were stained with DAPI and Phalloidin red overnight. The migration of epithelium was measured by scanning the corneas with ZEISS Axio-observerZ inverted microscope as described in as described in Materials and Methods.

**Figure 2 cells-13-01599-f002:**
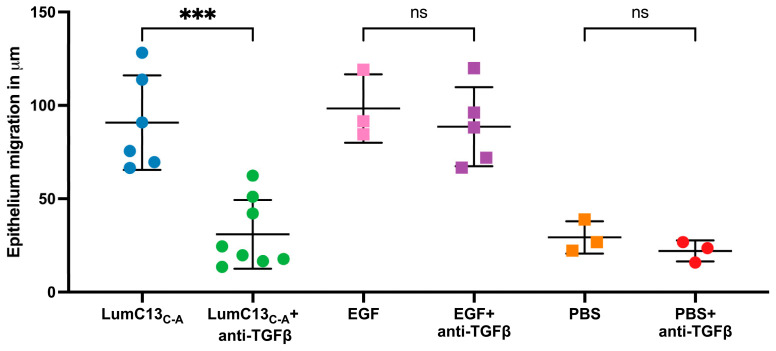
Cumulative data of the effects of anti-TGFβ antibodies on cornea epithelial migration in lumikine- and EGF-treated corneas. Cornea epithelium debridement and cornea epithelial migration of injured corneas were determined as described in Figure 1. Each cornea was scanned at four different quarters of the specimens and the mean epithelial migration of each individual cornea was calculated. Effects of EGF were not affected by anti-TGFβ antibodies. PBS alone has little effect on epithelium migration at 3 h. Each dot represents the mean of epithelium migration in each of individual corneas. Probability *p* value *** <0.001; ns, no significant difference.

**Figure 3 cells-13-01599-f003:**
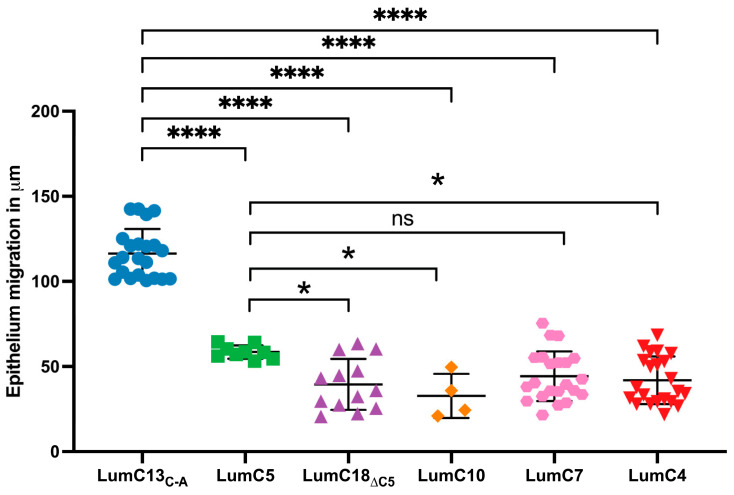
Identification of minimum and essential LumC peptides promoting healing of epithelial debridement. Adult 8–12-week-old *Lum^−/−^* KO mice were subjected to epithelium debridement with Algerbrush^®^. Ten microliters of eye drops containing various Lum C-terminal peptides (0.3 µM in PBS) were administered to the epithelium-debrided corneas every 10 min for 4 h, processed for whole mount and then subjected to scanning for corneal epithelium migration with ZEISS Axio-observerZ as described in Figure 1. The data show the effects of different LumC terminal peptides on epithelial migration, i.e., lumikine (LumC13_C-A_), LumC18_∆C5_ peptide missing the last five amino acids of C-terminal EVTLN, and shorter peptides missing the N-terminal amino acids, i.e., LumC10, LumC7, LumC5, and LumC4, failed to promote cornea epithelium migration. LumC5 has slightly higher capability in promoting epithelium than the rest of peptides missing the N-terminal amino acids except LumC7. Probability value: * <0.05, **** <0.0001; ns, no significant difference.

**Figure 4 cells-13-01599-f004:**
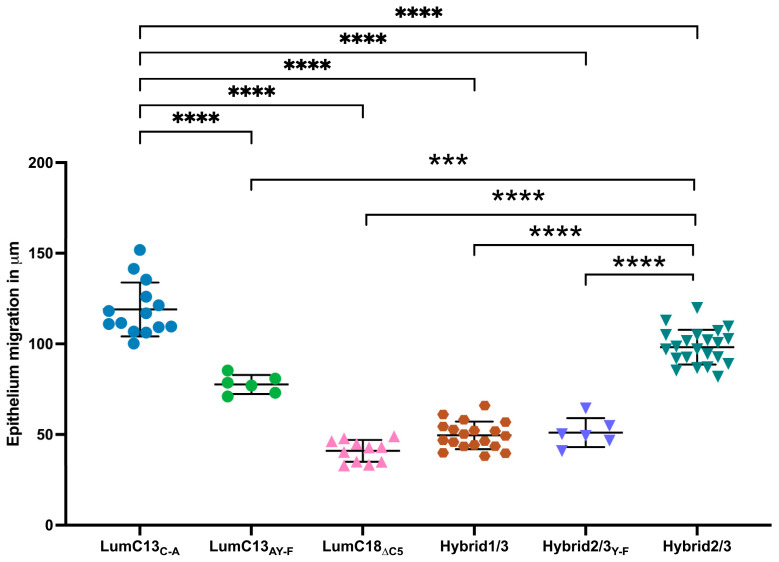
Role of N-terminal Y residue in lumikine activity in promoting corneal epithelium migration. Lum^−/−^ mice were used to determine the role of Y residue in lumikine function as described in Figure 1. The epithelium migration was evaluated in injured corneas treated with Hybrid1/3 peptide consisting of EVTLN but missing Y residue, and peptides in which Y is substituted by F, i.e., LumC13_C-A/Y-F_ and Hybrid2/3_Y-F_. Both missing Y and substitution of F greatly reduce the peptides’ capacity in promoting epithelium migration. Probability value: *** <0.001, **** <0.0001.

**Figure 5 cells-13-01599-f005:**
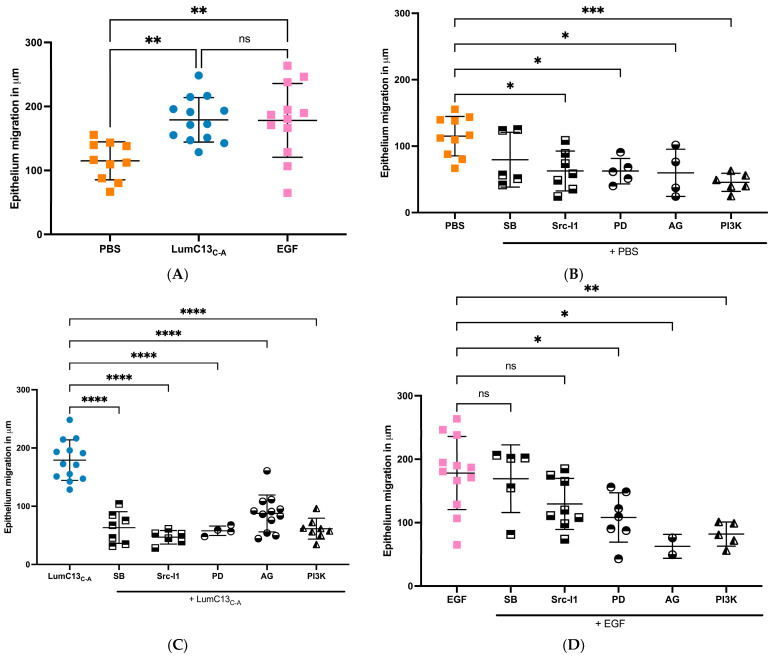
Effect of inhibitors of TGFβ receptor, EGF receptor, and MAPK signaling cascades on the epithelium migration promoted by lumikine. Experimental *Lum^−/−^* mice were subjected to epithelium debridement (2 mm in diameter) and allowed to heal for 5 h. The injured corneas were treated every 10 min with 10 µL eye drops containing LumC13_C-A_ (0.3 µM) and EGF (10 ng/mL) with inhibitors ALK5 inhibitor SB431542 (10 µM), EGFR inhibitor AG1478 (10 nM), pERK1/2 inhibitor (PD98059, 5 µM), PI3K inhibitor (wortmannin, 1 µM), and Src inhibitor (SrcI-1, 2 µM), respectively, as described in Figure 1 and Figure 2. The experimental mice were euthanized and the epithelium migration was determined by whole count scanning of excised corneas stained with phalloidin and DAPI. Images were taken with a Zeiss Apotome microscope (Observer Z1). (**A**): both LumC13_C-A_ and EGF promoted epithelial migration; (**B**): all inhibitors inhibited epithelium migration in PBS, except Alk5 inhibitor; (**C**): increased epithelium migration by LumC13_C-A_ decreased in the presence of all inhibitors; (**D**): increased epithelium migration by EGF was inhibited by Ag1478 (ERK inhibitor), AG (EGFR inhibitors) and Wortmann (PI3K inhibitor), but not SB431542 (Alk5 inhibitor) and SrcI-1 (Src inhibitor). Probability value: * <0.05, ** <0.01, *** <0.001, **** <0.0001; ns, no significant difference.

**Figure 6 cells-13-01599-f006:**
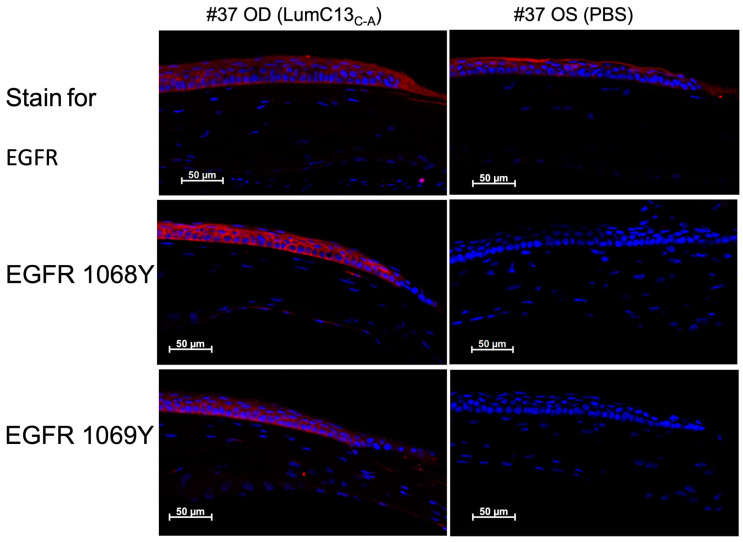
Administration of lumikine leads to EGFR activation. Epithelium-debrided corneas of *Lum^−/−^* mice were treated with lumikine and PBS eyedrops for 6 h. Cryosections of experimental corneas were then subjected to immunofluorescence staining with antibodies against EGFR, phospho-Y1068 and -Y1069 of EGFR as described in Materials and Methods.

**Figure 7 cells-13-01599-f007:**
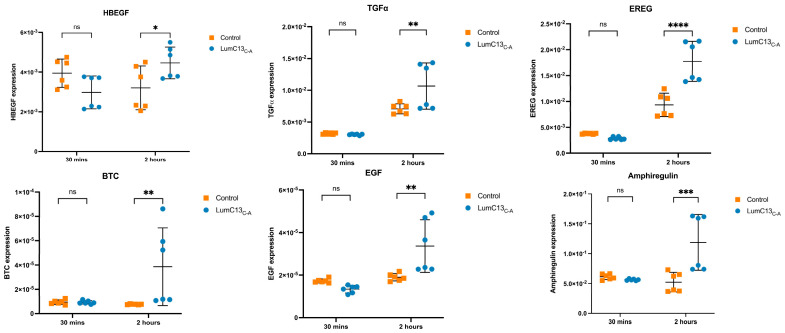
Expression of EGFR ligands by HTCE cells treated with lumikine. Cultures of human telomerase-immortalized corneal epithelial (HTCE) cells at late-log phase were scratched and incubated basic medium containing 0.3 µM lumikine for 30 min and 2 h, cells harvested were subjected to qRT-PCR for expression of various EGFR ligands, i.e., HBEGFE (heparin-binding EGF), TGFα, EREG (epiregulin), BTC (betacellulin), EGF, and AREG (amphiregulin) as described in Materials and Methods. In 30 min, none of the ligands’ expressions were upregulated. At 2 h expression of all ligands were upregulated. However, both BTS and EGF have low expression levels, whereas HBEGF, TGFα, EREG, and AREG expression was upregulated. Probability values: * <0.05, ** <0.01, *** <0.001, **** <0.0001; ns, no significant difference.

**Figure 8 cells-13-01599-f008:**
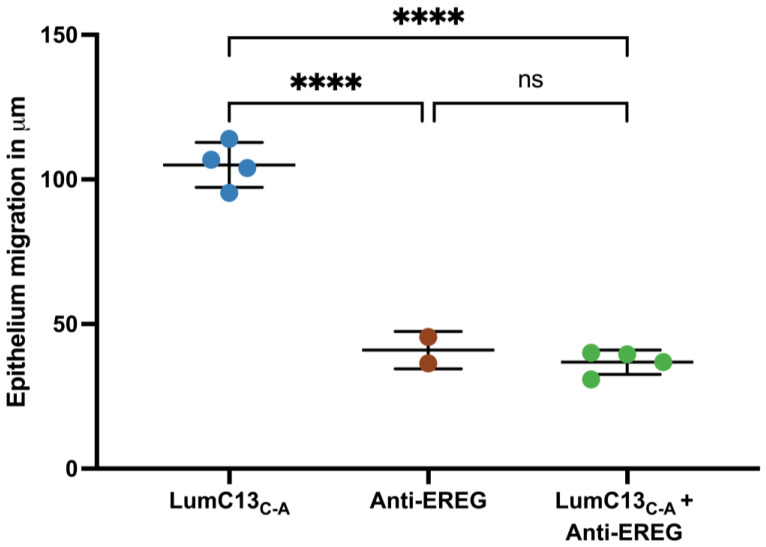
Effects of anti-EREG antibodies on migration of injured cornea epithelium treated with lumikine. Adult *Lumi^−/−^* (8–12 weeks old) mice were subjected to corneal epithelium debridement. The injured corneas were treated with eye drops containing lumikine (0.3 µM), lumikine (0.3 µM) + goat anti-mouse EREG antibodies (20 µg/mL), (R & D, CAT# AF10680). Probability values: **** <0.0001; ns, no significant difference.

**Figure 9 cells-13-01599-f009:**
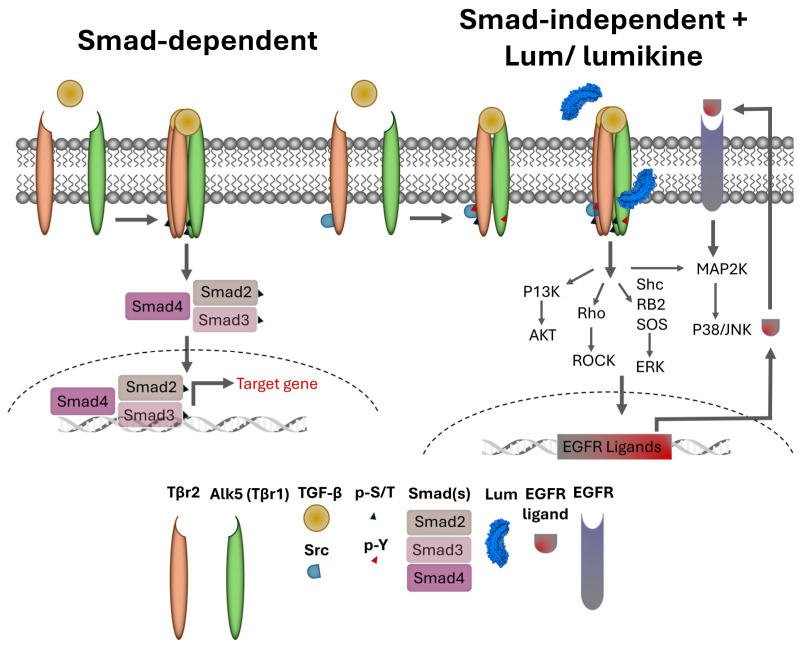
Roles of Lum/lumikine in TGFβ/TBR (TGFβ receptor) signaling in the healing of corneal epithelium debridement. There are two TGFβ/TBRs signaling pathways: (1) Canonical Smad-dependent pathway binding of TGFβ to TBR2 (type 2 receptor) initiates autophosphorylation of TBR2 and forms tetrameric TBRs (TBR2_2_/ALK5_2_) in which several serine and threonine residues in ALK5 (TBR1) are phosphorylated by p-TBR2. The activated tetrameric TBRs subsequently phosphorylate Smad 2 and 3 that bind Smad 4 to form a Smad 2/3/4 complex and translocate to nuclei, where it binds other transcription factors and drives expression of TGFβ target genes, e.g., components of extracellular matrix. (2) The noncanonical Smad-independent pathway is characterized by the binding of SRC to free TBR2; the subsequent TGFβ binding to TBR2 triggers SRC phosphorylates 284Y of TBR2 and form a tetrameric TBR_2_/ALK5_2_ complex that phosphorylates Alk5 Y residues and transduces the MAPK signaling cascades. The binding of free lumican (secreted by epithelial cells and/or keratocytes) and lumikine (LumC13_C-A_) to Alk5 lead to upregulated expression of EGFR ligands that switch on and feed forward the signaling cascades to p38MAPK/JNK and/or ERK pathways.

## Data Availability

The original contributions presented in the study are included in the article/Appendix A, further inquiries can be directed to the corresponding author.

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
