# Peer review of "Lumican/Lumikine Promotes Healing of Corneal Epithelium Debridement by Upregulation of EGFR Ligand Expression via Noncanonical Smad-Independent TGFβ/TBRs Signaling"

_cells, 2024, doi:10.3390/cells13191599_

Round 1

Reviewer 1 Report

Comments and Suggestions for Authors

In this study the authors examine the wound healing effects of a lumican peptides on epithelial debridement wounds in lumican-null mice to identify the minimum and essential stretch of amino acids in the terminal 13 aa long peptide that supports epithelial migration. They have identified that the terminal 5 aa of lumikine is needed but not sufficient to promote epithelial migration. They show that this activity involves SMAD-independent TGF beta signaling where it seems to promote adhesion of SRC. But this inference is not supported very well. Their data also suggest that EGFR signaling is promoted by Lumikine trough upregulation of EGFR ligands. Overall, this is a very nice study that provides molecular insights into the role of lumican in epithelial woundhealing.

The following clarifications would be helpful.

1)    Introduction -  please clarify that lumican binds interacts withbthe TLR signaling mechansims, but is=t does not directly bind TLR4. Rather it binds cell surface CD14, Cav 1 and also beta 2 integrin.

2)    Line 86 -  instead of Ref 33 which is not a review, I think the authors wanted to cite the review in Matrix Biol (PMID: 37793508)

3)    The naming of the LumC13C-A to lumikine is confusing. Why not name every peptide with the substitution as the authors did for LumC13C-A. Is the peptide containing the C terminal 13 aa named Lumikine or is the Lum13 where the C has been replaced with an A Lumikine?

4)    In Figure 1B, inclusion of one set with no peptide treatment at all would have been a good control.

5)    Line 137 should read epithelial migration.

6)    Fig. 3: Please state that Fig 4 shows the cumulative data for this experiment.

Comments on the Quality of English Language

Someone needs to edit the english and check for clarity.

Reviewer 2 Report

Comments and Suggestions for Authors

The work of Kao et al. investigates the role of a Lumican deprived-peptide in respect epithelial wound healing. The authors claim that a non-reducible form (LumC13c-a) activates wound healing in an corneal model and of wound healing in general. However, I wonder and can't understand this happens: A paper* of a similar group of authors has not been cited, in which the peptide LumC13c-a was already introduced and the main effect of the peptide was described. Under this circumstances, I don't feel comfortable to review this "new extended version".

*Sudhir Verma 1, Fernando T Ogata 2, Isabel Y Moreno 2, Cassio Prinholato da Silva 2, Tainah Dorina Marforio 3, Matteo Calvaresi 3, Mehmet Sen 4, Vivien J Coulson-Thomas 2, Tarsis Ferreira Gesteira. Rational design and synthesis of lumican stapled peptides for promoting corneal wound healing. Ocul Surf 2023 Oct:30:168-178. 

Reviewer 3 Report

Comments and Suggestions for Authors

Overview

This study investigates the role of Lumicans C-terminal 13 amino acids (LumC13) and its variant Lumikine in promoting corneal epithelial cell healing, specifically through the upregulation of EGFR ligand expression and noncanonical Smad-independent TGFβ/Tβr signaling pathways. The study includes both in vivo and in vitro experiments, utilizing Lum-/- mice and HTCE cells, demonstrating the importance of Lumikine in corneal epithelial cell migration.

Review Criteria

1. Scientific Merit

Strengths:

·The article designs detailed experiments to elucidate the role of Lumicans C-terminal in corneal epithelial healing.

·The use of the Lum-/- mouse model provides physiologically relevant data and the combination of in vivo and in vitro experiments enhances the credibility of the findings.

Weaknesses:

·The explanation of some mechanisms is somewhat vague, particularly concerning the noncanonical Smad-independent TGFβ signaling pathway.

· The description of the control groups in the experimental design is not detailed enough, especially compared to other known healing-promoting factors.

Suggestions:

·Provide a more detailed molecular mechanism explanation, especially regarding the noncanonical Smad-independent signaling pathway.

·Supplement detailed descriptions and statistical analyses of the control groups to ensure the reliability of the results.

2. Completeness

Strengths:

·The article is well-structured, with a clear progression from background introduction to experimental results, followed by discussion and conclusion.

·The experimental methods are comprehensive, allowing others to replicate the study.

Weaknesses:

·The discussion section is relatively brief and does not fully explain some experimental results.

·There is a lack of detailed discussion on potential applications and future research directions.

Suggestions:

·Expand the discussion section with more in-depth analysis and explanations of the experimental results, particularly for experiments that did not meet expectations.

·Include a discussion on the potential clinical applications of this research and future research directions.

3. Figures and Illustrations

Strengths:

·The figures effectively summarize the experimental data and results, aiding in the understanding of the study's findings.

Weaknesses:

·The color scheme used in the figures is not aesthetically pleasing, which might detract from the overall presentation of the data.

·The mechanism diagram appears to be roughly made, which may reduce its effectiveness in conveying complex biological interactions.

Suggestions:

·Select a cohesive and visually appealing color palette that enhances readability and clarity. Consider using software tools like ColorBrewer or Adobe Color for selecting harmonious color schemes suitable for scientific illustrations.

·Improve the clarity and detail of the mechanism diagram. Ensure that all components  are clearly labeled and visually distinct. Use appropriate symbols and icons to represent different elements and interactions.

Conclusion

Overall, this article is well-designed and the results are reliable, contributing significantly to the understanding of corneal healing mechanisms. However, there is room for improvement in the interpretation of results and readability. It is recommended that the authors revise and resubmit the article to enhance its scientific merit, completeness, and reader appeal.

Summary of Recommendations

1. Provide a more detailed explanation of the noncanonical Smad-independent signaling pathway mechanism.

2. Supplement detailed descriptions and statistical analyses of the control groups.

3. Expand the discussion section with more in-depth analysis and explanations of the experimental results.

4. Discuss the potential clinical applications and future research directions of this study.

5. Include brief explanations of key terms and simplify the description of the results section.

6. Select a visually appealing and cohesive color palette for the figures.

7. Use professional diagram tools to create high-quality, detailed mechanism diagrams.
